# The flagellar motor of *Vibrio alginolyticus* undergoes major structural remodeling during rotational switching

**Brittany L Carroll[1,2†], Tatsuro Nishikino[3†], Wangbiao Guo[1,2], Shiwei Zhu[1,2‡], Seiji Kojima[3], Michio Homma[3*], Jun Liu[1,2*]**

[1]Department of Microbial Pathogenesis, Yale School of Medicine, New Haven, United States; [2]Microbial Sciences Institute, Yale University, West Haven, United States; [3]Division of Biological Science, Graduate School of Science, Nagoya University, Furo-cho, Chikusa-ku, Nagoya, Japan

**\*For correspondence:**
g44416a@cc.nagoya-u.ac.jp (MH);
jliu@yale.edu (JL)

†These authors contributed equally to this work

**Present address:** ‡Howard Hughes Medical Institute, Yale School of Medicine, New Haven, United States

**Competing interests:** The authors declare that no competing interests exist.

**Abstract** The bacterial flagellar motor switches rotational direction between counterclockwise (CCW) and clockwise (CW) to direct the migration of the cell. The cytoplasmic ring (C-ring) of the motor, which is composed of FliG, FliM, and FliN, is known for controlling the rotational sense of the flagellum. However, the mechanism underlying rotational switching remains elusive. Here, we deployed cryo-electron tomography to visualize the C-ring in two rotational biased mutants in *Vibrio alginolyticus*. We determined the C-ring molecular architectures, providing novel insights into the mechanism of rotational switching. We report that the C-ring maintained 34-fold symmetry in both rotational senses, and the protein composition remained constant. The two structures show FliG conformational changes elicit a large conformational rearrangement of the rotor complex that coincides with rotational switching of the flagellum. FliM and FliN form a stable spiral-shaped base of the C-ring, likely stabilizing the C-ring during the conformational remodeling.

## Introduction

Many bacteria navigate complex environments by controlling the flagellar rotational switch between counterclockwise (CCW) and clockwise (CW). *Escherichia coli* and *Salmonella enterica* (henceforth, *Salmonella*) use a 'run-and-tumble' approach for controlling movement, by which flagella rotating in a CCW sense drive the cell body forward, and when the rotation sense switches to CW the bacterium tumbles through the medium to change direction (*Berg, 2003*; *Chevance and Hughes, 2008*; *Terashima et al., 2008*). *Vibrio alginolyticus* has a unique three-step swimming pattern with forward, reverse, and flick motions; where CCW rotation propels the cell body forward, CW rotation drives the bacterium in reverse, and a flicking motion occurs upon CW-CCW rotation to change swimming direction, analogous to the tumble (*Xie et al., 2011*).

The motor is the most intricate part of the flagellum, not only responsible for flagellar assembly and rotation but also essential for rotational switching. Spanning from the cytosol through the outer membrane, the motor consists of a series of rings, with the L-ring at the outer membrane, the P-ring located within the periplasmic space, the MS-ring embedded in the cytoplasmic membrane, and the C-ring inside the cytoplasm (*Homma et al., 1987*; *Francis et al., 1992*; *Ueno et al., 1992*; *Francis et al., 1994*). The stator subunits, embedded in the cytoplasmic membrane, generate torque via ion flow to rotate the C-ring (*Blair, 2003*; *Sato and Homma, 2000a*). Different from *E. coli* and *Salmonella*, *V. alginolyticus* possesses several Vibrio-specific features: H-, T-, and O-rings. The O-ring is located on the outside of the outer membrane (*Zhu et al., 2017*), the H-ring facilitates the outer membrane penetration of the flagellum (*Terashima et al., 2010*; *Zhu et al., 2018*), and the T-ring

contacts the H-ring and stators, presumably acting as a scaffold to hold the stators (*Terashima et al., 2006*; *Zhu et al., 2019*).

Flagellar rotation is powered by an electrochemical gradient across the cell membrane,driving ion flow through the stator complex (*Berg, 2003*; *Terashima et al., 2008*; *Kojima and Blair, 2004*; *Li et al., 2011*). In *Salmonella* and *E. coli*, $H^+$ ions are conducted through the stator units. In *Vibrio* species, $Na^+$ ions are conducted through the stator units. The prevailing idea is that MotA and MotB (in the $H^+$ motor) or PomA and PomB (in the $Na^+$ motor) form a membrane-bound stator subunit (*Sato and Homma, 2000a*; *Kojima and Blair, 2004*; *Braun et al., 2004*; *Sato and Homma, 2000b*). Recent cryo-EM studies have shown that MotA and MotB assemble in 5:2 stoichiometry (*Santiveri et al., 2020*; *Deme et al., 2020*). The A and B subunits have four and one transmembrane helices, respectively, and two helices from the A subunit and one from the B subunit form an ion channel that contains an essential ion-binding aspartyl residue (*Hosking et al., 2006*). Inactive stator complexes diffuse through the cytoplasmic membrane and interact with the rotor to generate a series of conformational changes within PomA and PomB that open the ion channel and facilitate binding to the peptidoglycan layer (*Hosking et al., 2006*; *Mino et al., 2019*; *Zhu et al., 2014*; *Fukuoka et al., 2009*; *Sudo et al., 2009*; *Kojima et al., 2018*). However, exactly how the stator assembles around the motor and interacts with the C-ring is still not well understood.

The C-ring is essential for flagellar rotation and rotational switching. The C-ring structure is conserved among diverse species, with repeating subunits consisting of FliG, FliM, and FliN, with FliN sometimes being supplemented or replaced by FliY (*Kojima and Blair, 2004*; *Zhao et al., 1995*). FliG, located closest to the cell membrane (henceforth referred to upper portion) of the C-ring, interacts with the MS-ring, the lower C-ring, and the stator via three domains (*Lee et al., 2010*). The interaction of the N-terminal domain of FliG ($FliG_N$) with FliF tethers the C-ring to the MS-ring and is necessary both for assembly and rotation of the flagellum (*Ogawa et al., 2015*; *Lynch et al., 2017*; *Xue et al., 2018*). The middle domain of FliG ($FliG_M$) interacts with FliM, holding the upper, membrane-proximal portion of the C-ring in contact with its lower, membrane-distal portion (*Brown et al., 2002*; *Brown et al., 2007*; *Minamino et al., 2011*). Lastly, the C-terminal domain of FliG ($FliG_C$) interacts with a cytoplasmic loop of PomA (or MotA in proton-driven motors) via interactions of oppositely charged residues, connecting the stator and rotor (*Lloyd and Blair, 1997*; *Yakushi et al., 2006*; *Takekawa et al., 2014*). FliM also has three domains (*Park et al., 2006*). The N-terminal domain of FliM ($FliM_N$) binds to the chemotaxis signaling protein phosphoryl CheY (CheY-P) to trigger switching from CCW to CW rotation (*Paul et al., 2011*; *Vartanian et al., 2012*). The $FliM_M$ serves as a connection between the base of the C-ring and FliG (*Brown et al., 2002*; *Minamino et al., 2011*). $FliM_C$ forms a heterodimer with FliN when fused via a flexible linker (*Notti et al., 2015*; *Dos Santos et al., 2018*). The third protein, FliN, a small single-domain protein, dimerizes with FliM or itself to form the base of the C-ring (*Brown et al., 2005*). Proposed models for FliG, FliM, and FliN assembly include 1:1:4 (*Sarkar et al., 2010a*; *Sarkar et al., 2010b*) or a 1:1:3 (*McDowell et al., 2016*) stoichiometry. Two FliN homodimers form a ring at the base of each C-ring subunit in the first model (*Sarkar et al., 2010a*), while a FliM:FliN heterodimer and a FliN homodimer create a spiral base of the C-ring in the second model (*McDowell et al., 2016*).

It has been proposed that FliG undergoes a dramatic conformational change from an open, extended form during CCW rotation to a closed, compact form during CW rotation (*Lee et al., 2010*). Two α-helices play an important role in determining the FliG conformation: helix$_{MN}$ connects $FliG_N$ and $FliG_M$, and helix$_{MC}$ connects $FliG_M$ and $FliG_C$. It was suggested that these helices are rigid and extended in the open CCW conformation, while they become disordered to switch into the compact CW conformation. Each domain contains armadillo (ARM) repeat motifs; $ARM_N$ interacts with the adjacent FliG monomer, and $ARM_M$ and $ARM_C$ interact either inter- or intramolecularly, depending upon the CCW or CW rotational sense, respectively (*Lee et al., 2010*; *Brown et al., 2002*; *Minamino et al., 2011*). The domain-swapping mechanism was proposed to coordinate with the conformational change in helix$_{MC}$ (*Lee et al., 2010*). Another biochemical study suggested that the domain swapping occurs during C-ring assembly, with FliG in solution existing as a monomer, and an equilibrium favoring FliG oligomers in the C-ring (*Baker et al., 2016*).

The FliG conformational change occurs about a hinge region first predicted by in silico modeling and characterized via mutational analysis (*Van Way et al., 2004*). Two recently characterized fliG variants in *V. alginolyticus* (G214S and G215A), previously characterized in *E. coli* (G194S and G195A [*Van Way et al., 2004*]), appear to hinder the conformational change sterically (*Nishikino et al.,*

2018; *Nishikino et al., 2016*), resulting in motors that rotate primarily in a single direction. The wild-type flagella rotate in a CW:CCW ratio of 1:3, while the fliG-G214S mutant produces a CCW-biased phenotype (CW:CCW 1:9), and fliG-G215A mutant produces a CW-locked phenotype (*Nishikino et al., 2016*). These adjacent residue substitutions create opposite motility phenotypes and are located in the Gly-Gly flexible linker, a hinge region whose conformation is believed to be dependent upon helix$_{MC}$ (*Nishikino et al., 2016*).

To characterize C-ring dynamics during switching, we used cryo-electron tomography (cryo-ET) to determine in situ motor structures of the *V. alginolyticus* CCW-biased mutant G214S and the CW-locked mutant G215A. We found that a large conformational change of the C-ring occurs between the CCW- and CW-motors. Docking the previously solved homologous structures into our cryo-ET maps, we built two models of the C-ring in CCW and CW rotation.

## Results

### Visualization of intact flagellar motors in CCW and CW rotation states

To address the mechanism of rotational switching, we utilized cryo-ET to visualize the C-rings of two FliG mutants (G214S and G215A) of *V. alginolyticus.* These mutants were previously shown to have CCW-biased and CW-locked flagellar motors, respectively (*Nishikino et al., 2018*). Each *V. alginolyticus* cell usually possesses a single, polar, sheathed flagellum, which limits the amount of information per cell for in situ motor structure determination. We therefore introduced the *fliG*-G214S and *fliG*-G215A mutations (*Table 1*) into the *flhG* KK148 strain, which produces multiple flagella at the cell pole (*Kusumoto et al., 2006*). Tomograms were collected using a 300kV Titan Krios electron microscope (Thermo Fisher) with Volta phase plate, a GIF energy filter and a post-GIF K2 detector electron detector (Gatan). Multiple flagella can be readily seen in a typical cryo-ET reconstruction (*Figure 1A,D*). In total, 2221 CCW-biased motors from G214S cells and 1618 CW-locked motors from G215A cells were used to determine in situ flagellar motor structures (*Figure 1B,E* and *Table 1*). Due to increased contrast from the Volta phase plate and focused classification, we observed 34-fold symmetry of the C-ring in both variants (*Figure 1C,F* and *Figure 1—figure supplement 1*). To begin to address potential differences in the switch complex, we measured the diameter of the C-ring in Fiji ImageJ (*Schindelin et al., 2012*). To describe the observed changes, we will refer to the top of the C-ring as the outermost region facing the cytoplasmic membrane. Starting from the top portion in the *fliG* G214S (CCW) the diameter is 46.2 nm, the middle 46.6 nm, and the bottom 49 nm (*Figure 1C,I*). In *fliG* G215A (CW) motor the diameters are 49.0 nm, 46.6 nm, and 48.3 nm, respectively (*Figure 1F,I*). Thus, the CW-motor is about 20.8 Å wider than the CCW-motor at the top of the C-ring, even though the number of C-ring subunits remains unchanged (*Figure 1*). To understand the interaction between the C-ring and the stator, we also analyzed the densities on the top of the C-ring. We were able to resolve the stator ring in the CW-locked motor but not in the CCW-biased motor (*Figure 1G,H*). Interestingly, the diameter of the stator ring is 48.3 nm, falling in between the CCW-biased motor and CW-locked motor diameters (*Figure 1H*).

To increase the resolution of the C-ring, we refined the C-ring structures with 34-fold symmetry applied to approximately 18 Å for the CCW and 19 Å for CW rotations (*Figure 2A,D*). The increased resolution allows us to reveal two unique conformations of the C-ring (*Figure 2*). We observed a significant lateral shift upon switching from CCW to CW, while the overall diameter of the C-ring varies slightly between the two mutants. These data demonstrate that the C-ring undergoes a large lateral conformational change during flagellar switching that leads to a difference in FliG presentation evident from the diameter change at the top of the C-ring. The protein composition of the C-ring likely remains the same as both CCW- and CW-motors have 34-fold symmetry, and the middle and bottom regions of the C-ring are similar in diameter.

**Table 1.** Cryo-ET data used in this study.

| Strain genotype | Tomograms | Motors |
| --- | --- | --- |
| *fliG-G214S* | 259 | 2221 |
| *fliG-G215A* | 220 | 1618 |

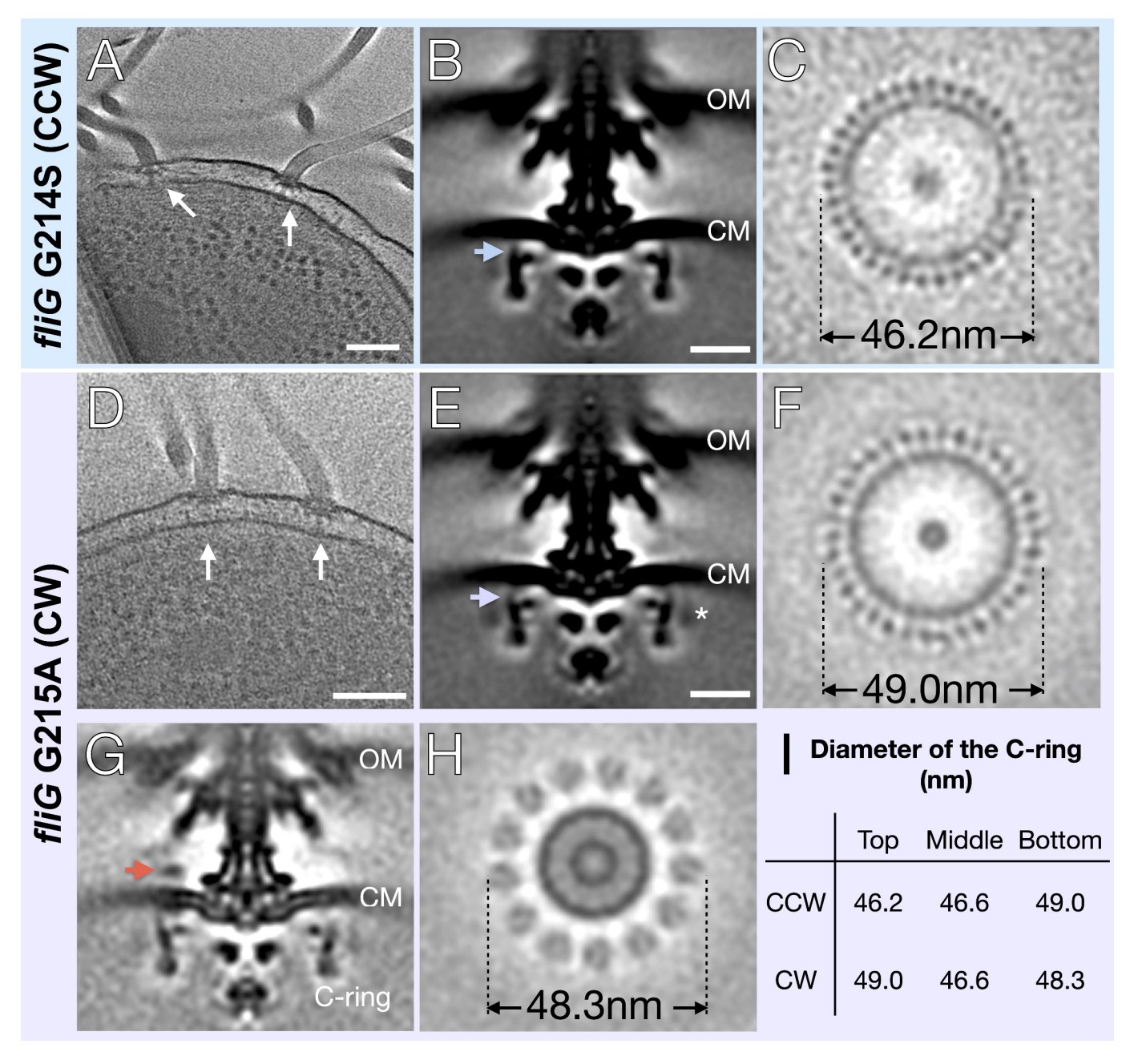

**Figure 1.** In situ structures of *V. alginolyticus* flagellum motor in CCW and CW rotation. (**A**) A representative tomogram slice showing two motors in *V. alginolyticus fliG* G214S mutant (white arrow) Scale bar 100 nm. (**B**) A medial cross-section of the in situ flagellar motor structure. Scale bar 25 nm. (**C**) A perpendicular cross-section of the motor showing the top portion of the C-ring (blue arrow in B). (**D**) A representative tomogram slice of the CW-motor in *V. alginolyticus fliG* G215A CW-locked variant. (white arrow) Scale bar 100 nm. (**E**) A medial cross-section of the in situ flagellar motor structure. Scale bar 25 nm. (**F**) A perpendicular cross-section of the motor showing the top portion of the C-ring (purple arrow in E). (**G**) A medial cross-section of the flagellum motor with density for the stator (orange arrow). (**H**) A perpendicular cross-section of the motor showing the stator ring (orange arrow in G). (**I**) Diameters of the C-ring measured at the top, middle, and bottom from perpendicular cross-sections for both the CCW- and CW-motors. Abbreviations: outer membrane (OM), cytoplasmic membrane (CM).

The online version of this article includes the following figure supplement(s) for figure 1:

**Figure supplement 1.** Classification of the CCW and CW flagellar motor structures reveals 34-fold symmetry of the C-ring.

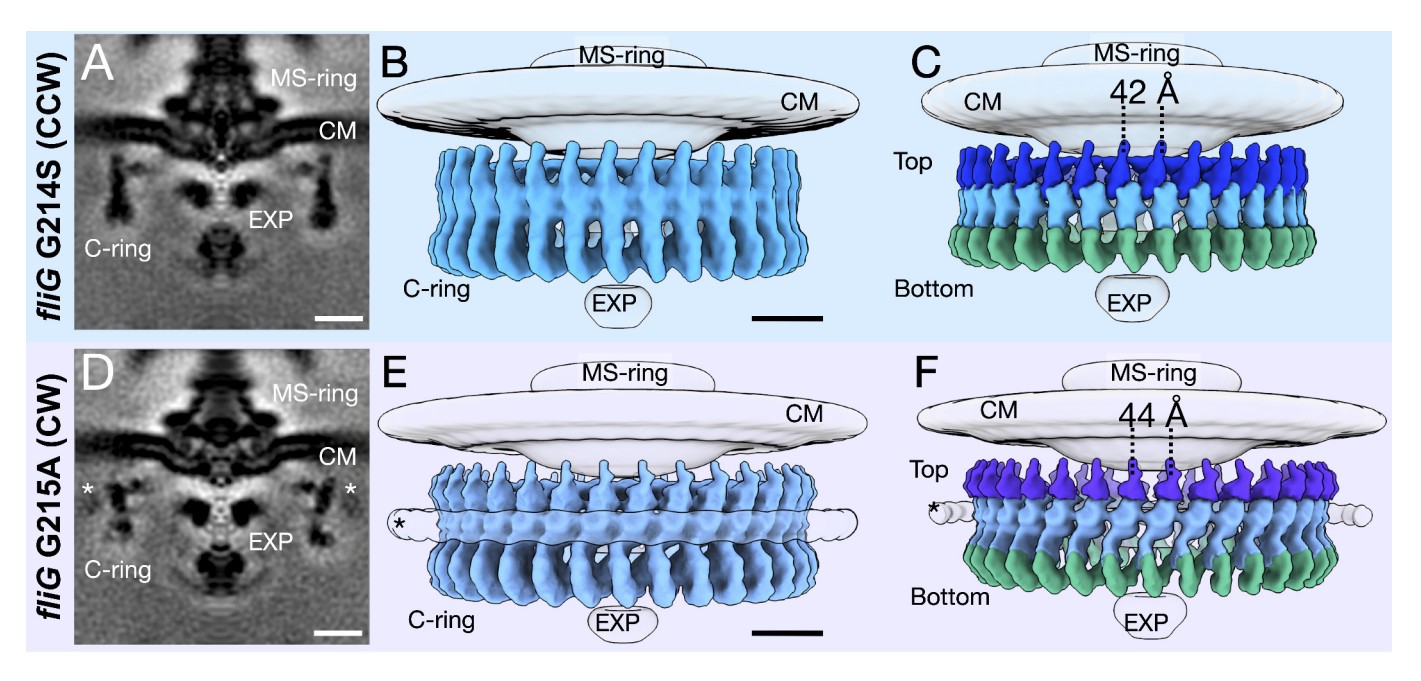

**Figure 2.** Focused refinement of the C-ring reveals differences in the CCW- and CW-motors. (**A**) A medial cross-section of the focused refinement of the C-ring in *V. alginolyticus fliG* G214S CCW-biased variant. Scale bar 25 nm. (**B**) A side view of the CCW C-ring. Scale bar 10 nm. (**C**) A side view of the segmented map shows the top (blue), middle (light blue), and bottom (green) regions. (**D**) A medial cross-section of the C-ring in *fliG* G215A CW-locked variant. Scale bar 25 nm. (**E**) A side view of the focused refined CW-locked C-ring. Scale bar 10 nm. (**F**) A side view of the focused refined C-ring further segmented into the top (purple), middle (gray blue) and bottom (green) regions of the C-ring. The asterisk highlights the additional density (white) observed in only the CW-motor that we speculate to be CheY-P. Abbreviations: cytoplasmic membrane (CM), export apparatus and ATPase (EXP).

## Molecular architectures of the CCW and CW C-rings

To understand the molecular detail, we built pseudo-atomic models of the C-ring by docking available crystal structures from the C-ring proteins into the in situ maps derived from sub-tomogram averaging. First, we used homologous structures deposited in the PDB and SCWRL4 (*Krivov et al., 2009*) to map the *V. alginolyticus* amino acid sequence onto the open (PDB-3HJL [*Lee et al., 2010*]) and closed (PDB-4FHR [*Vartanian et al., 2012*]) conformations of the FliG structure. Specifically, we used the full-length extended FliG with crystal packing from PDB-3HJL to build the CCW model. The monomer generated from crystal packing fits into the cryo-ET map surprisingly well. The compact FliG from the PDB-4FHR model was used for the CW model, with the FliG$_N$ domain from PDB-3HJL, as there are no other structures of FliG$_N$. Second, we used I-TASSER (*Roy et al., 2010*; *Yang et al., 2015*) to generate the structures of FliM and FliN. Third, the models were placed into the cryo-ET map using UCSF ChimeraX (*Goddard et al., 2018*). The known protein-protein interfaces were preserved during docking and while the unknown protein interface between the FliMN heterodimer and FliN homodimer was refined with the Rosetta protein-protein docking function (*Lyskov and Gray, 2008*) before being placed into the cryo-ET map. Phenix (*Afonine et al., 2013*) was used to optimize the placement and to assess the overall fit using rigid body refinement.

We built the C-ring subunit with FliG at the top, FliM in the middle, and three FliN molecules at the base (see *Figure 2C,F* for clarification), supporting the 1:1:3 (FliG, FliM, and FliN) model in lieu of the 1:1:4 model, as there is no additional density for a fourth FliN molecule. The 1:1:3 model suggests a FliM-FliN heterodimer interacts with the FliN homodimer characterized by mass spectroscopy, as had been modeled into a previously solved cryo-EM map (*McDowell et al., 2016*; *Thomas et al., 2006*). To understand the differences in the CCW and CW structures, we first built the CCW model into four adjacent subunit patch. From this patch model single subunit was real space rigid body refined with the CCW subunit having CCmask of 0.69, and CCbox of 0.83 and the CW subunit had a CCmask of 0.66 and CCbox of 0.82 (*Figure 3A,C*). We also generated maps from

the models using ChimeraX molmap at 18 Å for the CCW model and 19 Å for the CW model. The in silico generated maps exhibit features similar to the experimental map (*Figure 3—figure supplement 1*) and (*Figure 3—figure supplement 2*).

We expanded our subunit model to the whole C-ring by applying 34-fold symmetry (*Figure 3B, D*). Importantly, the expanded model fits into the experimental map without clashes, and preserves the architecture of the base observed in the maps. In silico CCW and CW maps from the entire

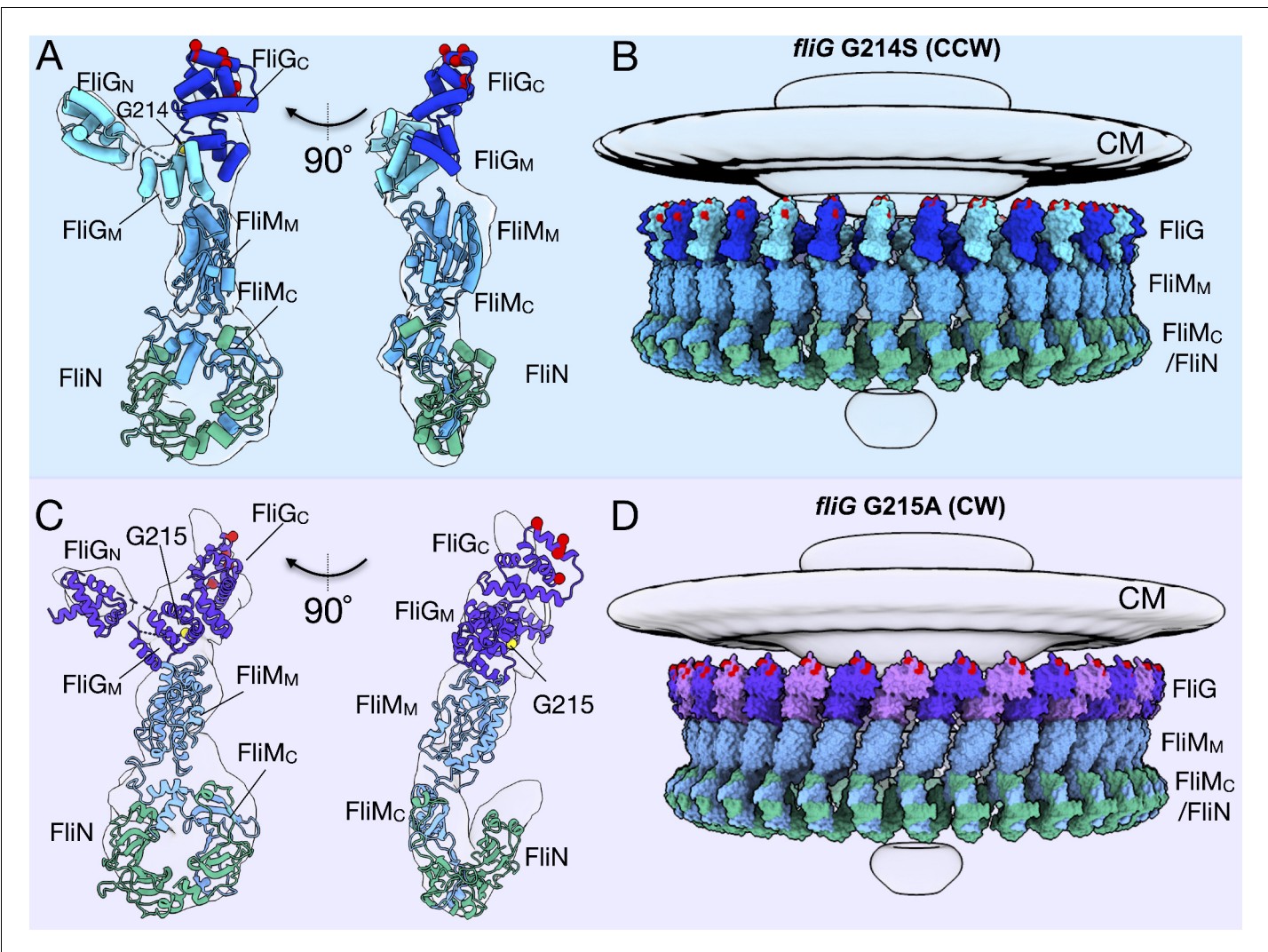

**Figure 3.** Molecular architectures of the C-ring in CCW and CW rotation. (A) Two views varying by 90° of a single C-ring subunit model fitted in the in situ map of *fliG* G214S (white). (B) A side view of the CCW-motor with the model shown as surface expanded for symmetry. The density map for the MS-ring, CM, and EXP are shown in gray for reference. FliG is alternating as dark blue, and aqua, FliM is colored light blue, and FliN is shown in green. The charged residues of FliG that interact with the stator are red spheres, and glycine 214 is shown as a yellow sphere. (C) Two views varying by 90° of a single C-ring subunit model in cartoon fitted into the cryo-ET density of *fliG* G215A (white). (D) A side view of the CW model rendered as surfaces expanded for symmetry. FliG is alternating as purple, and mauve, FliM is gray blue, and FliN is green. The charged residues of FliG that interact with the stator are red spheres, and glycine 215 is shown as a yellow sphere. The alternating coloring of FliG reflects the different crystal structures used to build the model. To build the CCW model, we used the extended FliG, PDB-3HJL, with crystal packing. To build the CW model, we used the compact FliG, in PDB-4FHR. These models support the domain swapping hypothesis. Abbreviations: cytoplasmic membrane (CM), export apparatus and ATPase (EXP).

The online version of this article includes the following figure supplement(s) for figure 3:

**Figure supplement 1.** In silico maps reflect similar features as experimental data.

**Figure supplement 2.** The important regions biochemically and structurally of the C-ring in the CCW and CW models.

**Figure supplement 3.** The FliG conformational change results in a tilt about FliM to change the presentation of FliG_C to the stator.

C-ring model were also generated using ChimeraX molmap, showing that the two models yield distinct maps similar to their respective rotational sense (*Figure 3—figure supplement 1*).

Our pseudo-atomic models can be further substantiated from the biochemical data. The ARM$_M$ (*Figure 3—figure supplement 2*, orange [*Park et al., 2006*]) and ARM$_C$ (*Figure 3—figure supplement 2*, green) motif orientations have been preserved from the crystal structures. The EHPQR motif (Glu144, His145, Pro146, Gln147, and Arg179, *Figure 3—figure supplement 2*, lime spheres) of FliG is located in ARM$_M$ and oriented to interact with FliM$_M$ (*Figure 3—figure supplement 2*, green sticks). Residues at the FliM-FliM interface had previously been identified in *Salmonella* and *E. coli* correspond to Asn56, His63, Asp184, Pro185, and Met187 in *V. alginolyticus* and they could interact with the neighboring FliM (*Figure 3—figure supplement 2*, magenta) (*Park et al., 2006*; *Sakai et al., 2019*). The N-terminus of FliM$_M$ is oriented facing outward toward the additional density of the CW-locked mutant in a position in which it can bind to CheY-P (*Figure 3—figure supplement 2*, asterisk and D) (*Lee et al., 2001a*). It is important to note that these residues were used to identify unique location in our model, but due to limited resolution their precise location is not known. Our models of the C-ring are thus consistent with both the previous biochemical data and our cryo-ET maps.

## FliM/FliN interactions hold the C-ring subunits together

The tilting observed between the CCW- and CW-motors is centered on FliM, which serves as a structural protein that holds the C-ring subunits together (*Figure 3*). However, without a full-length crystal structure, the relative orientations of FliM$_N$, FliM$_M$, and FliM$_C$ are unknown. The previous crystal structures and solution NMR studies showed FliM$_M$ interacting with FliG (*Vartanian et al., 2012*; *Dyer et al., 2009*; *Lam et al., 2013*). FliM$_C$ shows sequence homology to FliN, and structural homology to the FliN homodimer, as shown from the structure of a crystallized FliM$_C$-FliN fusion protein (*Notti et al., 2015*; *McDowell et al., 2016*). To place FliM into our map, two I-TASSER models, FliM$_M$ and FliM$_C$, and place the domains separately into our cryo-ET map, and then connected the linker regions (*Figure 3*). We lacked sufficient information to confidently place FliM$_N$, and we therefore left it out of our model. The position of FliM$_N$, which binds CheY-P, is flexible (discussed below) (*Lee et al., 2001b*). Our model shows that FliM$_M$ forms the middle portion of the C-ring and allows for the tilting of the C-ring, as FliM$_C$ tethers FliM to FliN via the heterodimer. We observed the same continuous helical density at the base of the C-ring as shown in *Salmonella* (*Thomas et al., 2006*) and proposed to be formed via alternating FliM$_C$-FliN heterodimers and FliN homodimers (*McDowell et al., 2016*) connecting adjacent subunits (*Figure 4A,B*). This region, known as a spiral (*McDowell et al., 2016*), is uniform within both the CCW and CW cryo-ET maps, and remains largely unaltered upon rotational switching (*Figure 4C*). We will refer to this as the FliM$_C$FliN3 (MCN3) spiral from here on. The expansion of the C-ring pseudo-atomic model preserves the continuous uniform shape (*Figure 4A,B*). By building the patch model initially we were able to preserve the similarity of the MCN3 spiral by removing potential bias from the segmentation of a monomer. The hydrophobic patch residues, originally identified in *Salmonella* (*McMurry et al., 2006*), that interact with FliH are facing toward the ATPase such that the interaction could occur (*Figure 4D*). It is important to note that there may be slight movements of the MCN3 proteins during switching, which we would need higher resolution to observe. Taken together, these data suggest that perhaps the MCN3 spiral is important for keeping the C-ring subunits connected during the conformational remodeling and rotation.

## Presentation of FliG$_C$ to the stator in the CCW- and CW-motors

To characterize the observed conformational change, we created a superposition of the CCW and CW C-ring subunits in the CCW- and CW-motors. Using UCSF Chimera (*Pettersen et al., 2004*) an axis was drawn through the center of the model, FliG$_{MC}$ shown in red, FliG$_{MC}$ and FliM$_M$ in green, and FliM$_M$ alone in orange. This revealed a 23 Å shift of FliG with a 37° tilt about FliM (*Figure 3—figure supplement 3*). FliG and FliM tilt 20° outward upon switching from CCW to CW rotation (*Figure 3—figure supplement 3*). Furthermore, comparison of just the axes shows that the movement characterizing the large conformational change can be attributed to FliM$_M$ (*Figure 3—figure supplement 3* insets). This suggests that, upon switching, FliG is presumably presented to the stator units very differently as a result of the rearrangement of FliM$_M$.

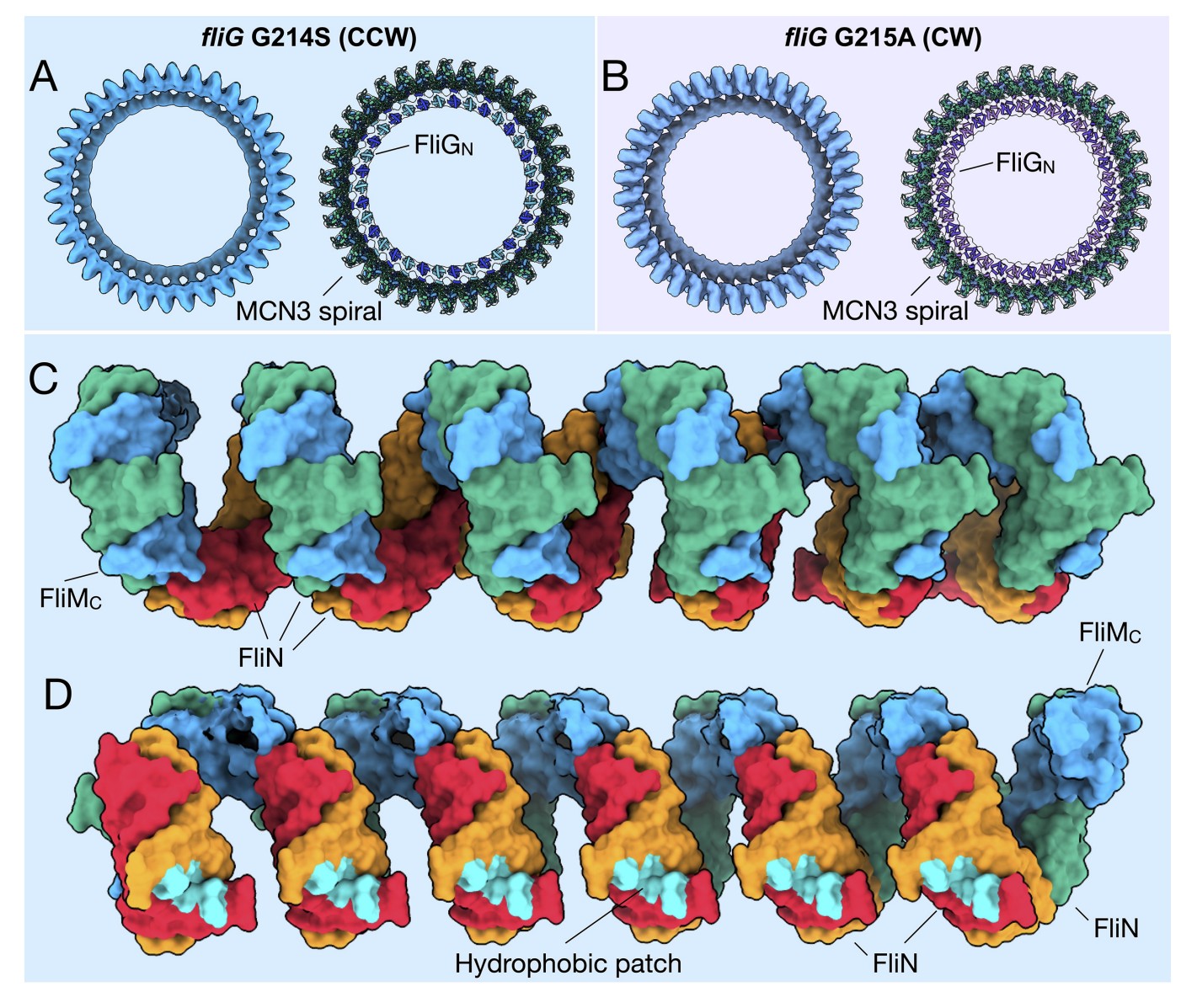

**Figure 4.** FliM$_C$ and FliN form a continuous spiral structure at the base of the C-ring. (**A**) A bottom view of the *fliG* G214S (CCW) motor map alone (light blue), and the model (cartoon) expanded for symmetry in the in situ map (gray transparent). (**B**) A bottom view of the *fliG* G215A (CW) motor map alone (gray blue), and the model (cartoon) expanded for symmetry in the in situ map (gray transparent). (**C**) A view from the outside periphery, and (**D**) a view from the center or inside looking out, are surface representations of six subunits that form a portion of MCN3 spiral from the CCW model, with one FliM$_C$ (light blue), and three FliN molecules (green, red, and orange). The hydrophobic patch of FliN (*McMurry et al., 2006*), L68, A93, V111, V113, Y118, that had been shown to interact of the FliH (cyan) points toward the center of the C-ring.

## Extra density around the CW C-ring

We observed an extra ring around the CW-locked C-ring (*Figure 2*). We speculate that it is formed by CheY-P, as there are several pieces of evidences in line with this model. First, the density is above the noise level in the CW-motors but not in the CCW-motors, which biochemically makes sense as CheY-P binds to CW rotating motors (*Paul et al., 2011*; *Vartanian et al., 2012*; *Dyer et al., 2009*; *Lee et al., 2001b*). The cells should have endogenous CheY, as there has been no alteration to the *CheY* gene. Second, density similar in location and shape has been shown two recent studies to be CheY-P. Chang et al. have used GFP-tagged CheY-P to show that, in *Borrelia burgdorferi*, CheY-P occupies the same position (*Chang et al., 2020*). Rossmann et al. also identified similar density for

the CheY homolog, CleD in Caulobacter crescentus (*Rossmann et al., 2020*). Third, when we place the crystal structure of CheY into the model, there is about 60 Å gap between the C-ring density and this additional density (*Figure 3—figure supplement 2*). This distance may be bridged by the 42 residues of FliM$_N$ to connect CheY-P to FliM$_M$.

## Discussion

The flagellar motor structures we determined provide direct evidence that the C-ring possesses two distinct conformations in CCW and CW rotation. The dynamics of the C-ring appear to be confined to the upper two-thirds of the structure, with the spiral base that connects the C-ring subunits remaining relatively static. We observed a 37° lateral and 20° medial tilting of the C-ring (*Figure 3—figure supplement 3*), which almost certainly alters its interactions with the stator. In our pseudo-atomic model, the charged residues (Lys284, Arg301, Asp308, and Asp309 in FliG$_C$) that are known to interact with the stator are located at the top of the C-ring. FliG$_C$ moves about 40 Å laterally (*Video 1*). This large remodeling of the C-ring presumably enables distinct interactions between the stator and the C-ring (*Video 2*). FliG$_{CN}$ contains the ARM$_C$ motif that interacts with FliG$_M$ and FliM$_M$, and FliG$_{CC}$ contains the charged residues that interact with the stator. A flexible linker attaching the two domains would allow for a range of movement in FliG$_{CC}$ relative to FliG$_{CN}$ and FliG$_M$, thus suggesting that the presentation of the charged residues varies greatly depending upon the conformation of FliG.

Notably, our cryo-ET model shows 34-fold symmetry for both the CCW and CW rotating C-rings, with only a modest diameter change in FliG. It has previously been hypothesized, based on experiments utilizing fluorescently tagged FliM and FliN, that the FliG, FliM, and FliN composition of the C-ring changes during rotational switching (*Delalez et al., 2014*; *Delalez et al., 2010*; *Hosu and Berg, 2018*; *Lele et al., 2012*). A recent single-particle cryo-EM study suggested that inter-subunit spacing and interactions of the C-ring account for motor switching in *Salmonella* (*Sakai et al., 2019*). That study reported a 9 Å decrease in C-ring diameter upon switching to CW. We report identical diameters at the middle of the C-ring in both rotational directions. However, we observed ~21 Å increase in the diameter and ~40 Å lateral change at FliG$_C$ of the CW-motor. In a parallel study, Chang et al. used a *B. burgdorferi* with a CheX deletion or CheY3 deletion to investigate the switching mechanism of spirochetes (*Chang et al., 2020*). Similar conformational changes of the C-ring were observed in both species; however, the stator complexes were resolved in *B. burgdorferi,* as they appear to be less dynamic than those in *V. alginolyticus*. Visualization of the stator complex provided direct evidence that the conformational change of the C-ring alters the rotor-stator interaction. The model of stator-rotor interactions controlling the rotational direction of the motor is further bolstered by the recent high-resolution cryo-EM models of MotAB (*Santiveri et al., 2020*; *Deme et al., 2020*). *Santiveri et al., 2020* further postulated that MotA rotates CCW in *Campylobacter jejunie* by showing that the conformations of wildtype MotAB and MotAB trapped in the protonated state are nearly identical, thus ruling out a dramatic conformational change in the stator unit. Therefore, we conclude that the change in rotational direction

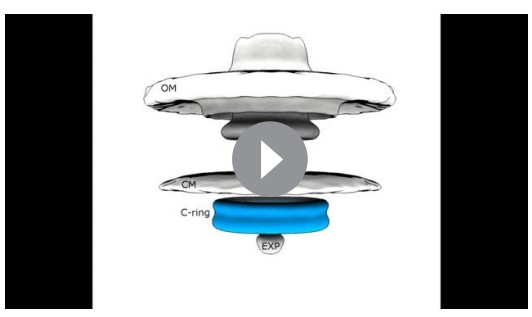

**Video 1.** Focused refinement of the C-ring reveals conformational change during rotational switching. The video starts with a *fliG* G214S motor, a mutation leads to a strong preference for CCW rotation, without symmetry applied. The C-ring is highlighted in blue. The video then zooms in to a focused refined C-ring, with much higher resolution, and turns in the CCW (looking down from the flagellum) direction. The pseudo-atomic model appears as cartoon in the map, as the map fades the model turns into surface rendering. Transparent density, that is likely CheY-P, binds and triggers the conformational change of rotor into the CW architecture. The video toggles between CCW and CW pseudo-atomic models in surface rendering to visualize the global changes of the C-ring architecture. It ends with the model spinning in the CW direction. The color scheme is the same as the paper. The video was rendered using ChimeraX.
https://elifesciences.org/articles/61446#video1

**Table 2.** Strains and plasmids.

| Strains or plasmids | Genotype or description | Reference or source |
|---|---|---|
| *V. alginolyticus* | | |
| VIO5 | Wild-type strain of a polar flagellum (Rif$^+$ Pof$^+$ Laf$^-$) | *Okunishi et al., 1996* |
| KK148 | VIO5 *flhG* (Multi-Pof$^+$) | *Kusumoto et al., 2008* |
| NMB328 | KK148 Δ*fliG* (Pof$^-$) | This study |
| *E. coli* | | |
| DH5α | Host for cloning experiments | *Grant et al., 1990* |
| S17-1 | *recA hsdR thi pro ara RP-4 2-tc::Mu-Km::Tn7* (Tp$^r$ Sm$^r$) | *Simon et al., 1983* |
| β3914 | *β2163 gyrA462 zei-298::Tn10* (Km$^r$ Em$^r$ Tc$^r$) | *Le Roux et al., 2007* |
| Plasmids | | |
| pMMB206 | Cm$^r$, P$_{tac}$P$_{lac}$UV5 | *Morales et al., 1991* |
| pNT1 | *fliG* in pMMB206 | *Takekawa et al., 2014* |
| pSW7848 | Suicide plasmid for allele exchange | *Val et al., 2012* |
| pHIDA3 | SacI fragment of *fliF-fliG* (55–175 internal deletion of FliG) in pSW7848 | *Mino et al., 2019* |

occurs from a large movement of the charged residues of FliG$_C$ relative to the stator.

Our in situ structures support that FliG, FliM, and FliN interact in a 1:1:3 ratio, as suggested for the non-flagellar homolog Spa33 in *Shigella* (*McDowell et al., 2016*). This stoichiometry favors the model in which FliM holds the individual C-ring subunits together by interacting with FliG$_M$ and forming a heterodimer with FliN. The FliM-FliN spiral connects adjacent C-ring subunits, allowing for dramatic movement without dissociation of the subunits. The FliG$_M$-FliM$_M$ interface, perhaps dynamic in its own right, has been suggested to be involved in flagellar switching (*Sakai et al., 2019*; *Dyer et al., 2009*; *Pandini et al., 2016*). It was shown, using NMR, that the CheY-P binding to FliM displaces FliG$_C$, and that the dramatic rearrangement of FliG$_{MC}$ is possible because of the flexibility of FliG (*Dyer et al., 2009*). In particular, the GGPG loop in FliM$_M$ is suggested to be critical for rotation of FliG$_M$ relative to FliM$_M$ (*Pandini et al., 2016*). Most recently, point mutations targeting FliM$_M$ were shown to restore CCW rotation in a CW-biased mutant (*Sakai et al., 2019*). These results, combined with our findings, lead us to suggest that the conformational change in FliG results in a rotation about FliM$_M$ that leads to the tilting of the C-ring subunit to alter the presentation of the charged residues of FliG$_C$ to the stator.

Given that the putative CheY-P-ring is likely associated with the CW C-ring (*Figure 1*), we propose a CCW-to-CW switching model (*Figure 5*, *Videos 1* and *3*). CheY-P binding to the FliM results in a conformational change in FliG and alters the interactions between the C-ring and stator. FliM$_C$ and FliN create a spiral base that holds the C-ring together during these dynamic rearrangements. FliM$_C$ and FliN may also be involved in relaying the conformational change to adjacent C-ring subunits. By placing the FliG crystal structures of the open and closed variations within our cryo-ET image, we provide additional evidence to support the proposed model in which the ARM$_M$ and ARM$_C$ domains toggle between inter- and intramolecular interactions (*Lee et al., 2010*). NMR data suggest that FliG$_C$ is the domain that moves and that FliG$_M$ remains in stable contact with FliM (*Dyer et al., 2009*). These changes in the C-ring structure produce the two directions of flagellar rotation. Understanding the interactions of the C-ring with the stator and the MS-ring is essential to elucidate the mechanism of rotational switching and transmission of the rotation to the flagellar filament. We could not resolve the detailed interactions between the cytoplasmic portion of the stator and the C-ring in both rotational directions, and we could not confirm directly that the orientation of FliG$_{CC}$ alters the presentation of its charged residues to the charged residues of PomA. Restricting the movement of FliG$_{CC}$ via truncations of the flexible linker may address the importance of FliG$_{CC}$ mobility. With more data and further classification and refinement of the CCW-biased motor, we expect to explain the FliG$_{CC}$ and stator interactions in great detail. With the rapid development of

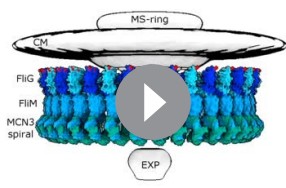

**Video 2.** C-ring rearrangement leads to altered FliG<sub>C</sub> presentation to the stator. The video starts with a side view of the CCW pseudo-atomic model in surface rendering, and the view rotates so that we are looking down on the C-ring from the flagellum. We can see the charged residues of FliG<sub>CC</sub> that interact with the stator. The video toggles between CCW and CW models, and asterisks appear to mark the charged residues of every other FliG. This highlights the large change,~40 Å, in FliG upon switching. A morphing of four adjacent subunits shows the dramatic movement of FliG<sub>C</sub> relative to the stator from the top view, and side view, the CCW model is shown as transparent. The video ends with the CW pseudo-atomic model. The video was rendered using ChimeraX.

https://elifesciences.org/articles/61446#video2

cryo-ET, it is increasingly possible to reveal motor structure *in situ* at higher resolution, which will further our understanding of the flagellar assembly and function.

In summary, we have used cryo-ET to visualize a major conformational change of the C-ring during rotational switching due to a single point mutant in FliG. Molecular modeling attributes the tilt component of the conformational change to FliM. Gyration of FliG about FliM presents the charged residues of FliG to the stator in a manner that controls the rotational sense. These movements within the C-ring subunits may be relayed throughout the switch complex by interactions between FliM and FliN within the spiral.

## Materials and methods

### Bacterial strains, plasmids, and growth condition

Bacterial strains used in this study are listed in *Table 2*. To introduce the *fliG* deletion, NMB328 was constructed from KK148 using pHIDA3 by allelic exchange as previous reported (*Kusumoto et al., 2006*; *Le Roux et al., 2007*). *V. alginolyticus* strains were cultured at 30℃ on VC medium (0.5% [wt/vol] polypeptone, 0.5% [wt/vol] yeast extract, 3% [wt/vol] NaCl, 0.4% [wt/vol] $K_2HPO_4$, 0.2% [wt/vol] glucose) or VPG medium (1%[wt/vol] polypeptone, 3% [wt/vol] NaCl, 0.4% [wt/vol] $K_2HPO_4$, 0.5% [wt/vol] glycerol). If needed, chloramphenicol and Isopropyl β-D-1-thio-galactopyranoside (IPTG) were added at final concentrations of 2.5 μg/ml and 1 mM, respectively. *E. coli* was cultured at 37℃ in LB medium (1% [wt/vol] Bacto tryptone, 0.5% [wt/vol] yeast extract, 0.5% [wt/vol] NaCl). When culturing *E. coli* β3914 strain, 2,6-diaminopimelic acid was added to the LB medium to a final concentration of 300 μM. If needed, chloramphenicol was added at final concentrations of 25 μg/ml.

### Mutagenesis

To introduce mutations (G214S or G215A) in the *fliG* gene on plasmid pNT1 site-directed mutagenesis was performed using the QuikChange method, as described by the manufacturer (Stratagene). All constructs were confirmed by DNA sequencing. Transformation of *V. alginolyticus* with plasmid pNT1 was performed by conjugational transfer from *E. coli* S17-1, as described previously (*Okunishi et al., 1996*).

### Sample preparation

The methods of sample preparation, data collection, data processing, and sub-tomogram analysis were followed as described previously (*Zhu et al., 2018*). *V. alginolyticus* cells were cultured overnight at 30℃ on VC medium, diluted 100 × with fresh VPG medium, and cultured at 30℃. After 4 or 5 hr, cells were collected and washed twice and finally diluted with TMN500 medium (50 mM Tris-HCl at pH 7.5, 5 mM glucose, 5 mM MgCl, and 500 mM NaCl). Colloidal gold solution (10 nm diameter) was added to the diluted *Vibrio* sp. samples to yield a 10 × dilution and then deposited on a freshly glow-discharged, holey carbon grid for 1 min. The grid was blotted with filter paper and rapidly plunge-frozen in liquid ethane.

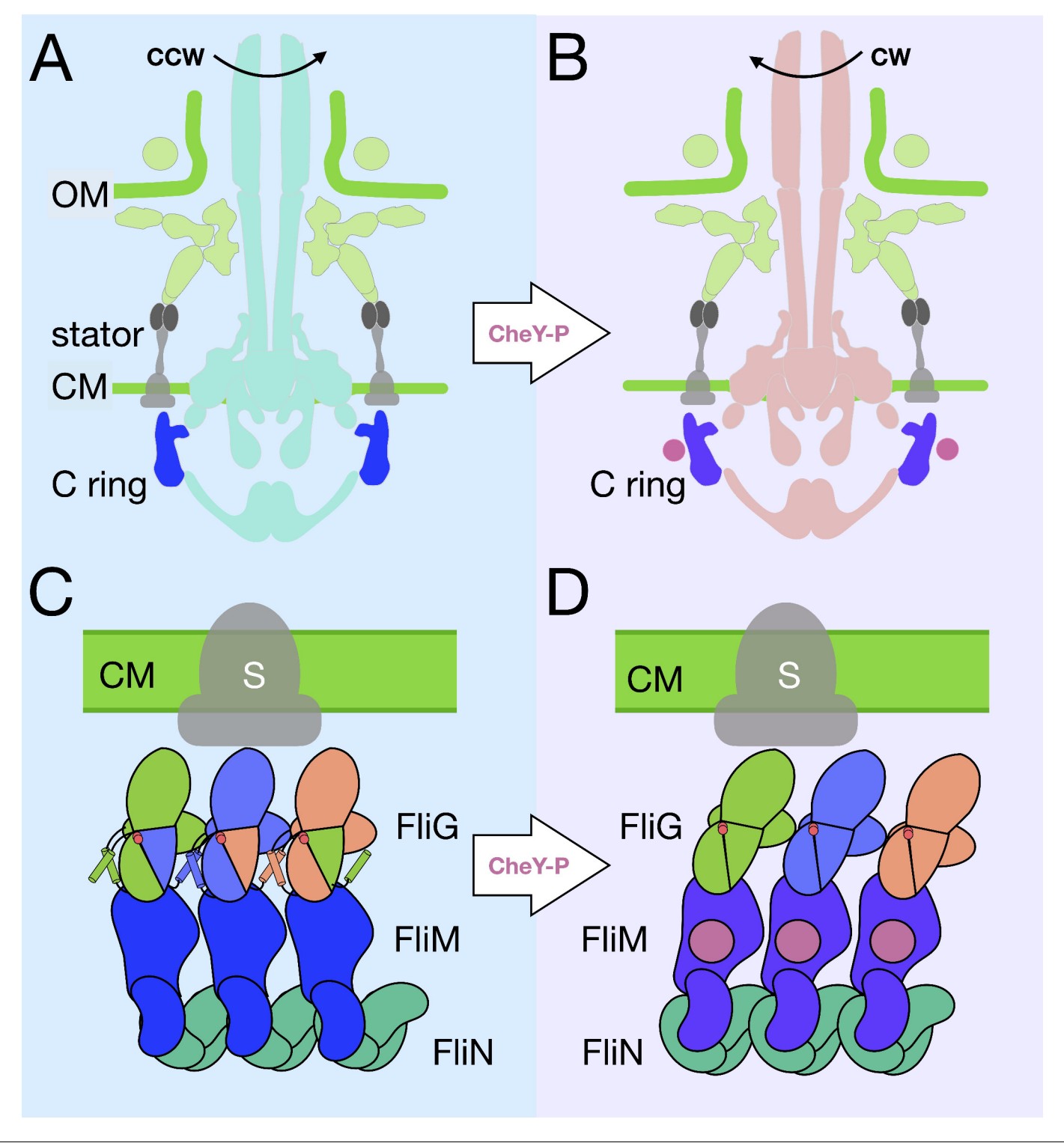

**Figure 5.** A model for the rotational switching. (**A**) A cartoon of the intact CCW-motor with the C-ring highlighted in blue, and the other conserved regions that have been resolved in teal, the *Vibrio*-specific proteins in light green, PomAB is colored gray as it has yet to be resolved in detail in *Vibrio*. (**B**) A cartoon of the intact motor that is rotating CW, C-ring is colored purple, the other conserved structures in coral, PomAB in gray, and CheY-P in pink. (**C**) A cartoon representation of the C-ring rearrangement upon CheY-P binding and rotational switching to the CW sense. In CCW rotation, the intermolecular interactions of FliG $ARM_M$ and $ARM_C$ are possible because $helix_{NM}$ and $helix_{MC}$ are ordered. The Gly-Gly flexible region is depicted by red circles. The stator (gray) is shown to interact with $FliG_C$. Upon CheY-P binding, the C-ring undergoes a conformational change that produces CW

*Figure 5 continued on next page*

*Figure 5 continued*

rotation. During this transition, the center of mass tilts laterally and slightly outward. In CW rotation there are intramolecular interactions of FliG ARM$_M$ and ARM$_C$. CheY-P (dark purple) is shown bound to FliM, and the stator (gray) interacts with FliG$_C$. We hypothesize that the conformational change in FliG is initiated *in vivo* by CheY-P binding, and this switches the rotational direction of the C-ring by changing how FliG$_C$ interacts with the stator. FliG undergoes a conformational change and FliM tilts about the base of the spiral. The spiral ensures that the C-ring subunits are connected while FliG and FliM undergoes large conformational changes.

## Data collection and processing

The frozen-hydrated specimens of NMB328 was transferred to a Titan Krios electron microscope (Thermo Fisher Scientific). The microscope is equipped with a 300-kV field emission gun (FEG), a GIF energy filter, and a post-GIF K2 Summit direct electron detector (Gatan). The images were collected at a defocus near to 0 μm using a Volta phase plate and the energy filter with a 20 eV slit. The data were acquired automatically with SerialEM software (*Mastronarde, 2005*). For better data collection, the phase shift is normally distributed in the range of 0.33π to 0.67π. A total dose of 50 e⁻/Å (*Chevance and Hughes, 2008*) was distributed among 35 tilt images covering angles from −51° to +51° at tilt steps of 3°. For every single tilt series collection, the dose-fractionated mode was used to generate 8 to 10 frames per projection image. Collected dose-fractionated data were first subjected to the motion correction program to generate drift-corrected stack files (*Li et al., 2013*). The stack files were aligned using gold fiducial markers and volumes reconstructed using IMOD and Tomo3d, respectively (*Kremer et al., 1996*; *Agulleiro and Fernandez, 2015*). In total, 259 tomograms of CCW-state motor (G214S mutation) and 220 tomograms of CW-state motor (G215A mutation) were generated.

## Sub-tomogram analysis with I3 package

Bacterial flagellar motors were detected manually, using the I3 program (*Winkler, 2007*; *Winkler et al., 2009*). We selected two points on each motor, one point at the C-ring region and another near the flagellar hook. The orientation and geographic coordinates of selected particles were estimated. In total, 1618 and 2221 sub-tomograms of *V. alginolyticus* motors from CW-motor and CCW-motor, respectively, were used for sub-tomogram analysis. The I3 tomographic package was used on the basis of the 'alignment by classification' method with missing wedge compensation for generating the averaged structure of the motor, as described previously (*Zhu et al., 2017*). To resolve the symmetry of the C-ring, we used focused refinement to the C-ring density. We then used multivariate statistical analysis for 3D classification, which allowed us to determine the 34-fold symmetry of the C-ring. It's important to point out that we did not resolve the 34-fold symmetry by imposing different symmetries. Furthermore, the C-ring structures in the two states share the same symmetry, however they have different conformations. We did similar analysis on the stator region to resolve 13 stator units in CW-motors. However, we were unable to resolve the stator units in CCW-motors. The reason behind the difference is currently unknown.

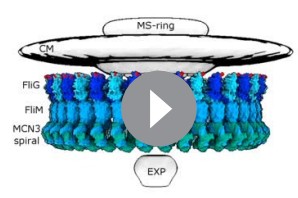

**Video 3.** A model for the rotational switching. The psudeo-atomic CCW model in surface rendering side view is rotated such we are looking up the motor toward the flagellum. The MCN3 spiral is in view, and the hydrophobic patch of FliN that interact with FliH is highlighted. The view then rotates back through the side view to top view, where the charged residues of FliG$_C$ are highlighted. The view returns to side view and zooms into a patch of four adjacent subunits. The CCW subunits morph into the CW architecture, highlight the large lateral change about FliM, and the domain swapping occurs as the FliG ARM$_{MC}$ interactions change from intramolecular to intermolecular. The MCN3 spiral exhibits little change. The video ends with the CW psudeo-atomic model. The video was rendered using ChimeraX.

https://elifesciences.org/articles/61446#video3

## Model generation

The *V. alginolyticus* C-ring proteins were generated using I-TASSER version 5.1 (*Yang et al., 2015*). The proteins were trimmed to the homologous structures deposited in the PDB to avoid large clashes. Using I-TASSER we have generated two models of *Vibrio* FliM, FliM full length, and FliM$_C$ with the linker region before. The full-length FliM generated by I-TASSER had the correct topology for FliM$_M$, but the orientation FliM$_C$ relative to FliM$_M$ was incorrect. Furthermore, the folding of FliM$_C$ is similar to the previously solved crystal structure, but different enough that the FliM-FliN heterodimer was unable to be modeled. To circumvent this problem, we ran I-TASSER with just the last 223 residues of FliM, including the FliM$_C$ and the flexible region immediately before. Due to the unknown relative location of the flexible regions it was necessary to trim. FliG, FliM, and FliN were hand guided by the literature into segmented patches of the CCW-biased and CW-locked cryo-ET maps corresponding to four subunits, and fit using the ChimeraX (*Goddard et al., 2018*) fit to map function. Mainly, four PDB models were used the full-length FliG (3HJL [*Lee et al., 2010*]), FliG-FliM (4FHR [*Vartanian et al., 2012*]), FliM-FliN fusion (4YXB [*Notti et al., 2015*]), and the FliN-dimer (1YAB [*Brown et al., 2005*]).

## Model refinement

The patch models were refined using PHENIX-1.17.1 Real Space Refinement (*Afonine et al., 2013*) to move the protein domains relative to one another while preserving the known architecture of the C-ring subunits. The unknown protein-protein interfaces were refined in Rosetta_2019.35 using the protein-protein docking scripts (*Lyskov and Gray, 2008*). The optimized single subunit model was then rigid body refined into a single subunit segmentation of the corresponding C-ring tomography map using PHENIX Real Space Refinement.

## Acknowledgements

We thank Michael Manson for critical reading and suggestions. We thank A Abe in our laboratory for technical support in this research. We thank Shenping Wu for her support on Krios. This work was supported in part by JSPS KAKENHI Grant Numbers JP16H04774 and JP18K19293 (to SK), JP18K06155 (to TK), and Program for leading Graduate Schools of Japan, Science for the Promotion of Science (JP17J11237 to TN). TN was supported in part by the Integrative Graduate Education and Research program of Nagoya University. BLC, SZ and JL were supported by grants GM107629 and R01AI087946 from National Institutes of Health. Molecular graphics and analyses were performed with UCSF ChimeraX, developed by the Resource for Biocomputing, Visualization, and Informatics at the University of California, San Francisco, with support from National Institutes of Health R01-GM129325 and the Office of Cyber Infrastructure and Computational Biology, National Institute of Allergy and Infectious Diseases.

## Additional information

### Funding

| Funder | Grant reference number | Author |
| --- | --- | --- |
| Japan Society for the Promotion of Science | JP16H04774 | Seiji Kojima |
| Japan Society of Ultrasonics in Medicine | JP18K19293 | Seiji Kojima |
| National Institute of Allergy and Infectious Diseases | AI087946 | Jun Liu |
| National Institute of General Medical Sciences | GM107629 | Brittany L Carroll Shiwei Zhu Jun Liu |
| Japan Society for the Promotion of Science | JP18K06155 | Tatsuro Nishikino |
| National Institutes of Health | R01AI087946 | Seiji Kojima |

| Japan Society for the Promotion of Science | JP17J11237 | Brittany L Carroll<br>Shiwei Zhu<br>Jun Liu |
| Japan Society for the Promotion of Science | JP17J11237 | Tatsuro Nishikino |

The funders had no role in study design, data collection and interpretation, or the decision to submit the work for publication.

## Author contributions
Brittany L Carroll, Formal analysis, Investigation, Visualization, Methodology, Writing - original draft; Tatsuro Nishikino, Conceptualization, Data curation, Formal analysis, Investigation, Writing - original draft, Writing - review and editing; Wangbiao Guo, Formal analysis; Shiwei Zhu, Data curation, Formal analysis; Seiji Kojima, Conceptualization, Supervision, Funding acquisition, Investigation, Writing - review and editing; Michio Homma, Conceptualization, Formal analysis, Supervision, Writing - original draft, Writing - review and editing; Jun Liu, Conceptualization, Supervision, Funding acquisition, Validation, Investigation, Writing - original draft, Writing - review and editing

## Author ORCIDs
Seiji Kojima (iD) http://orcid.org/0000-0002-5582-8935
Michio Homma (iD) https://orcid.org/0000-0002-5371-001X
Jun Liu (iD) https://orcid.org/0000-0003-3108-6735

## Decision letter and Author response
Decision letter https://doi.org/10.7554/eLife.61446.sa1
Author response https://doi.org/10.7554/eLife.61446.sa2

# Additional files

## Supplementary files
• Transparent reporting form

## Data availability
The resulting structures have been deposited in EMDB under accession codes EMD-21819 and EMD-21837.

The following datasets were generated:

| Author(s) | Year | Dataset title | Dataset URL | Database and Identifier |
| --- | --- | --- | --- | --- |
| Carroll BL | 2020 | Structures from: The flagellar motor of Vibrio alginolyticus undergoes major structural remodeling during rotational switching | https://www.ebi.ac.uk/pdbe/entry/emdb/EMD-21819 | Electron Microscopy Data Bank, 21819 |
| Carroll BL | 2020 | Structures from: The flagellar motor of Vibrio alginolyticus undergoes major structural remodeling during rotational switching | https://www.ebi.ac.uk/pdbe/entry/emdb/EMD-21837 | Electron Microscopy Data Bank, 21837 |

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
