## [Decision Letter]

[Editors’ note: the authors submitted for reconsideration following the decision after peer review. What follows is the decision letter after the first round of review.]

Thank you for submitting your work entitled "The flagellar motor of *Vibrio alginolyticus* undergoes major structural remodeling during rotational switching" for consideration by *eLife*. Your article has been reviewed by four peer reviewers, one of whom is a member of our Board of Reviewing Editors, and the evaluation has been overseen by a Senior Editor. The following individuals involved in review of your submission have agreed to reveal their identity: Chi Aizawa (Reviewer #2); Susan M Lea (Reviewer #3); Richard M Berry (Reviewer #4).

Our decision has been reached after consultation between the reviewers. Based on these discussions and the individual reviews below, we regret to inform you that your work will not be considered further for publication in *eLife* at this stage.

All reviewers agreed that an atomic model of the C-ring that allowed a molecular explanation for switching would be of great interest. The key issue is whether there is enough information in the tomograms to uniquely position existing atomic structures to enable this question to be addressed, and whether one can exclude significant conformational changes within these structures. At present the data shown do not allow the uniqueness and quality of the fit to be addressed. This paper is also unusual in that no estimates of resolution are provided. A further concern was the preprint of Chang et al. recently posted on bioRxiv. Is it possible that the mechanism proposed in Chang et al. only applies to *Borrelia burgdorferi*, while the mechanism in the present paper only applies to *Vibrio alginolyticus*? If this is true, that should have been stated in both papers, but it is stated in neither. In summary, all reviewers felt that a substantially revised paper that addressed all of these concerns should be considered again for publication in *eLife*.

Reviewer #1:

I had a difficult time judging the significance of the work. In some ways, the paper comes across as an incremental advance over previous studies on the bacterial flagellar motor. My main concerns had to do with the cryo-ET image analysis. This may be the first cryo-EM paper I have ever read where there is neither any mention of, nor discussion of, the resolution the reconstructed volumes. It is hard to understand how this can be absent, and it is difficult to understand the reliability of the models without some discussion of this point. Given the recent paper (Johnson et al., 2020) showing symmetry variation in the MS-ring complex, one wonders how reliable the overall imposition of a C34 symmetry is in the present reconstruction. There is no analysis of possible variance within these volumes, nor is there any information on what happens when other symmetries are imposed, both locally and globally. This is important, as the authors take one of the conclusions of their work to be that the composition of the C-ring remains constant between the two states. It is not clear how reliable such a conclusion can be at the limited resolution with the same imposed symmetry for both volumes.

Reviewer #2:

This is a beautiful work on conformational changes of the C ring of the flagellar motor.

The authors showed two conformations of the C ring in which rotation is biased to either CCW or CW.

Although I did not understand the details of the image processing using many sophisticated software, the results obtained are quite reasonable and convincing. I highly recommend this work to be published.

Reviewer #3:

The molecular details underlying how the bacterial flagellar motor switches direction of rotation is a long-studied question. This paper aims to address this question by using sub-tomogram averaging to generate volumes from directionally locked forms of the *V. alginolyticus* C-ring. This is an important question, being at the heart of how many bacteria move in response to their environment, and new data are therefore welcome. However, there are some issues to do with how the experiments are reported in light of past work and critical details of interpretation of the data that need to be made more explicit to allow a fuller understanding of how the new data enhance our current models. We would like to thank the authors for making the two volumes available as part of the review package to help with assessing the work.

Key prior work that provides the background to the current manuscript:

1) Prior structural work on intact C-rings by De Rosier and colleagues (Thomas et al., 2006) led to a moderate resolution, C34 reconstruction, of the C-ring with sufficient detail that permitted fitting of a spiral of a FliM_N_ homologue at the base of the ring (Mc Dowell et al., 2016). No formal resolution estimate was provided for the earlier reconstruction, but fitting of domains of known structure in a robust fashion was possible, at least at the bases of the C-ring.

2) Sakai et al., 2019, reported low resolution analyses of locked C-ring motors from *Salmonella* and demonstrate that there is only a minor change in diameter at the mid-point of the C-ring (~10A).

3) We (Johnson et al., 2020) have recently reported molecular detail for the MS ring on which the C-ring is assembled showing that, at least in the context of the isolated MS-ring, assemblies with different symmetries ranging from C32-C35 exist. This tallied with earlier analyses of C-ring stoichiometry, including Sakai (Sakai et al., 2019).

Novelty of these data:

These volumes are clearly the highest resolution views of the C-ring in locked CW and CCW states to date, combined with atomic structures of component proteins this gives a great opportunity to try and build a more detailed model of the complete C-ring in both states.

Major issues to be addressed:

1) Hand of the reconstructions? There are no details on how this key step was performed.

2) Symmetry of the assembly – there is no detail on how the tomograms were treated to assess the correct symmetry to apply, with only a simple statement on p10 that “our cryo-ET model shows 34-fold symmetry for both the CCW and CW rotating C-rings”. Full details, ideally including images, on how symmetry was assigned must be provided and evidenced in the manuscript. This is especially important in light of the previous data showing variation in C-ring symmetry as, in the extreme, the relatively subtle differences between the CW and CCW volumes could reflect different proportions of 34mer in the samples being averaged rather than true differences between the states.

3) Fitting of coordinates – it is not possible to judge the quality and uniqueness of the fit of coordinates to the volumes in the current version of the manuscript. A single correlation coefficient statistic is given in the text for each volume and it is unclear (1) if this is for the full model or simply the FliM_N_ spiral and (2) how each domain of the model contributes to this statistic. The figures that do show the fit show substantial extrusion of secondary structural elements outside of the volume and unfilled regions within the volume. This is perhaps unsurprising for a volume at modest resolution, but given that the authors go on to propose a radical structural rearrangement and then discuss the location of specific amino acids, the evidence for the uniqueness of the fit needs to be provided. Inspection of the volumes did not trivially allow the viewer to see a correlation in shape between the densities ascribed to the crucial FliG domains in the figures. There is also some confusion surrounding the FliM_c_/FliN spiral, as the text states that this portion of the structure is largely unaltered between the two states, but Figure 2 and Figure 2—figure supplement 1 show significant differences in the side views, with what looks like a 90 degree rotation of the spiral. How is this justified given the relatively featureless resolution of the maps?

4) Discussion of novelty throughout should acknowledge the incremental nature of the current findings c.f. the earlier work. In particular the extensive discussion c.f. the work of Sakai should acknowledge that, at the resolution of both studies, both could be summarized as “no major difference in diameter on switching”.

5) CheY binding – this is the most speculative part of the manuscript as a ring of density around the CCW ring is assigned as CheY-P largely on the basis of the groups work that will be presented in another manuscript (cited as Chang et al. in preparation). Particularly in light of another manuscript recently reported in bioRxiv that demonstrates the N-terminus of FliM is *not* required for CheY-P binding induced switching (Eisenback et al.), the authors should either provide further supporting experiments or more clearly mark this as speculative.

Reviewer #4:

The paper presents insitu structures of the bacterial flagellar motor in CW and CCW states, and a corresponding detailed ("pseudoatomic") structural model of each state. Models are consistent with previous biochemical data on the proximity of various amino acid residues of the constituent proteins.

The work is novel and important, and of sufficient general interest for publication in *eLife*: it allows analysis of the structural nature of the flagellar switch, which has been a canonical example of a macromolecular switch since the discovery of flagellar switching in the 1970s.

1) There is a lack of clarity, perhaps even an apparent internal contradiction, as to whether the structure at the filament-distal end of the motor (the FliM_C_/ FliN_3_ super-helix, hereafter "MCN3 helix") is different between CW and CCW structures. This is an important point, worthy even of inclusion in the Abstract, as any changes in this part of the C-ring caused by the 2 point mutations in FliG would indicate long-range effects of very small mutational changes. *Specifically*: the MCN3 helix seems to be very different between CW and CCW in Figures 2C/D and Figure 2—figure supplement 1 B/D – with fairly substantial rearrangement of protein chains. But the density maps for CW and CCW in Figure 2A/B look very similar, and the lack of changes in the MCN3 helix is confirmed in Figure 4 and in the text. Is it possible that the repeating structure of 34 units has been cut differently between Figures 2C and D, and also Figures 2—figure supplement 1B and D? Or alternatively, that the outwardly similar shape of the MCN3 helix conceals a very different internal arrangement of protein chains? Or something else? This apparent contradiction needs to be addressed and removed. A pseudoatomic model allows and demands this level of inspection, with appropriate caveats about the confidence in the model itself.

2) Closely related to comment 1, it is hard to match corresponding parts of the CW and CCW models – particularly in Figures 2C/D and Figure 2—figure supplement 1B/D. In reality there must be a pathway for switching between the CW and CCW structures. The authors should explore the transition between the two, and supply one or more animations to illustrate the conclusions that they can draw from their model regarding this important point – which is the main result of the paper. Such animations would be a powerful illustration, clarification and explanation of Figures 2-4 (and perhaps also Figure 5). They would serve 3 important purposes:

a) Identify corresponding parts, including perhaps the possibility that the repeating structure has been cut differently to isolate the units displayed in Figure 2 (see comment 1).

b) Illustrate either a smooth pathway between CW and CCW states (i.e. requiring no significant unfolding of secondary structure to avoid steric clashes, as proposed in the FliG model of Lee et al., 2010); or the lack of one, indicating rather the need for substantial disruption of interactions during a switch. The current best model for the switch is that of conformational spread (e.g. Bai et al., Science 2010), in which an energetic penalty for boundaries between domains of CW and CCW structure is a key parameter. This paper might be able to add structural data to that model.

c) Highlight and illustrate changes in the orientation and location of FliG_c_, which are the "output" of the switch and most relevant in terms of understanding how it can cause a change in rotation direction upon interaction with the stator. These are discussed in the text, but it is hard to see them in the existing figures.

3) Regarding the extra density peripheral to the C-ring in the CW state, visible in Figure 1 and suggested to be CheY later in the paper (Figure 2—figure supplement 1, subsection “Extra diffuse density around the CW C-ring”). The authors should cite references that proposed that FliM_N_ is an unstructured "fishing rod" that serves to bind and concentrate CheY, which their hypothesis and data support. Also the alternative explanation (see e.g. Figure 7 in J Mol Biol. 430(22):4557-4579.) that this density could be extra copies of FliM, outside the ordered C-ring. This would reconcile previous observations of substantial recruitment of FliM to the C-ring (although in *E. coli*) during CW rotation with the lack of expansion of the 34-fold ring observed here.

4) Figure 5 legend is wrong – D and C are described as if they were parts of B and A respectively. More importantly, the figure presents a model where the domain-swap (visible in B as swapping of colour-labelled domains of each protein chain into the neighbouring structural unit) is present in one directional state (CCW, B) but not the other (CW, D). This may be a misunderstanding of the domain-swap model of the flagellar switch. Alteration of the domain-swap during switching is not necessary, nor is it proposed by Baker et al., 2016 (as incorrectly stated in the Introduction). In fact, quite the opposite: the model proposes that the ARM-ARM interface is very strong, the main interaction holding the FliG ring together, and therefore very unlikely to open except under the special conditions that prevail during assembly of the FliG ring. Instead, the cartoon of the CCW state, Figure 5B, appears entirely consistent with the model of Lee et al., 2010, and the switch to a domain-swapped alternative of Figure 5D would then be a data-supported refinement and confirmation of the model of Lee et al., 2010. Alternative models propose non-domain swapped structures (like the cartoon of Figure 5D) for both CW and CCW. Perhaps the authors could comment on whether their data support one or other model.

[Editors’ note: further revisions were suggested prior to acceptance, as described below.]

Thank you for resubmitting your work entitled "The flagellar motor of *Vibrio alginolyticus* undergoes major structural remodeling during rotational switching" for further consideration by *eLife*. Your revised article has been evaluated by John Kuriyan (Senior Editor), a Reviewing Editor, and three reviewers.

The manuscript has been improved but there are is a remaining issue that needs to be addressed before acceptance, as outlined below:

In general, the authors have addressed the concerns raised in the previous review. But the decision letter for the previous manuscript stated: "A further concern was the preprint of Chang et al. recently posted on bioRxiv. Is it possible that the mechanism proposed in Chang et al. only applies to *Borrelia burgdorferi*, while the mechanism in the present paper only applies to *Vibrio alginolyticus*? If this is true, that should have been stated in both papers, but it is stated in neither." This is not answered either in the response nor in the revised paper.

---

## [Author Response]

[Editors’ note: the authors resubmitted a revised version of the paper for consideration. What follows is the authors’ response to the first round of review.]

Reviewer #1:I had a difficult time judging the significance of the work. In some ways, the paper comes across as an incremental advance over previous studies on the bacterial flagellar motor. My main concerns had to do with the cryo-ET image analysis. This may be the first cryo-EM paper I have ever read where there is neither any mention of, nor discussion of, the resolution the reconstructed volumes. It is hard to understand how this can be absent, and it is difficult to understand the reliability of the models without some discussion of this point.

We thank the reviewer for providing critical comments on our insufficient descriptions on cryo-ET methods and the resulting structures. We have addressed the concerns. We hope that the importance of this work is better clarified.

Given the recent paper (Johnson et al., 2020) showing symmetry variation in the MS-ring complex, one wonders how reliable the overall imposition of a C34 symmetry is in the present reconstruction. There is no analysis of possible variance within these volumes, nor is there any information on what happens when other symmetries are imposed, both locally and globally. This is important, as the authors take one of the conclusions of their work to be that the composition of the C-ring remains constant between the two states. It is not clear how reliable such a conclusion can be at the limited resolution with the same imposed symmetry for both volumes.

We agree with the reviewer that the symmetry is important for this study. We have been studying the *Vibrio* motor for many years and published several papers including Zhu et al., 2017. We could not resolve the symmetry of the C-ring until this study. To overcome the limitation, we first used a relatively new technique (Volta phase plate) to collect cryo-ET data, resulting in reconstructions with better contrast. We then used multivariate statistical analysis for 3D classification, which allowed us to determine the 34-fold symmetry of the C-ring. It’s important to point out that we did not resolve the 34-fold symmetry by imposing different symmetries. Furthermore, the C-ring structures in the two states share the same symmetry, however they have different conformations. Again the 34-fold symmetric features were observed first before we impose the symmetry to increase the resolution.

Reviewer #2:This is a beautiful work on conformational changes of the C ring of the flagellar motor.The authors showed two conformations of the C ring in which rotation is biased to either CCW or CW.Although I did not understand the details of the image processing using many sophisticated software, the results obtained are quite reasonable and convincing. I highly recommend this work to be published.

We thank the reviewer for the positive comments. We have provided more details in the Materials and methods. Hopefully the revised manuscript is easier to understand.

Reviewer #3:The molecular details underlying how the bacterial flagellar motor switches direction of rotation is a long-studied question. This paper aims to address this question by using sub-tomogram averaging to generate volumes from directionally locked forms of the V. alginolyticus C-ring. This is an important question, being at the heart of how many bacteria move in response to their environment, and new data are therefore welcome. However, there are some issues to do with how the experiments are reported in light of past work and critical details of interpretation of the data that need to be made more explicit to allow a fuller understanding of how the new data enhance our current models. We would like to thank the authors for making the two volumes available as part of the review package to help with assessing the work.Key prior work that provides the background to the current manuscript:1) Prior structural work on intact C-rings by De Rosier and colleagues (Thomas et al., 2006) led to a moderate resolution, C34 reconstruction, of the C-ring with sufficient detail that permitted fitting of a spiral of a FliM_N_ homologue at the base of the ring (McDowell et al., 2016). No formal resolution estimate was provided for the earlier reconstruction, but fitting of domains of known structure in a robust fashion was possible, at least at the bases of the C-ring.2) Sakai et al., 2019, reported low resolution analyses of locked C-ring motors from Salmonella and demonstrate that there is only a minor change in diameter at the mid-point of the C-ring (~10A).3) We (Johnson et al., 2020) have recently reported molecular detail for the MS ring on which the C-ring is assembled showing that, at least in the context of the isolated MS-ring, assemblies with different symmetries ranging from C32-C35 exist. This tallied with earlier analyses of C-ring stoichiometry, including Sakai (Sakai et al., 2019).Novelty of these data:These volumes are clearly the highest resolution views of the C-ring in locked CW and CCW states to date, combined with atomic structures of component proteins this gives a great opportunity to try and build a more detailed model of the complete C-ring in both states.

We thank the reviewers for reading and providing insightful feedback. We appreciated their understanding of the importance of the work. We have revised the manuscript to incorporate the key prior work and highlight our major findings in this study.

Major issues to be addressed:1) Hand of the reconstructions? There are no details on how this key step was performed.

This is a critical concern. It is challenging to directly determine the handedness of cryo-ET reconstructions. We have three pieces of data to support the handedness of the reconstructions in this study. First, we have independently determined two structures of the C-rings. The base of the C-ring shares similar right-handed spiral structure in both states. Importantly, the right-handed spiral structure matches well with the crystal structure of a FliN homolog (Mc Dowell et al., 2016). Second, the exact same cryo-ET protocol was used to generate 3D reconstructions of periplasmic flagella in spirochetes. Consistent with previous findings, periplasmic flagella form a right-handed ribbon in our reconstructions. Third, our recent in situ structures of HIV-1 Env spikes match well with near-atomic structures of purified Env spikes by cryo-EM single particle analysis. Together, we believe that our reconstructions in this study have correct handedness.

2) Symmetry of the assembly – there is no detail on how the tomograms were treated to assess the correct symmetry to apply, with only a simple statement on p10 that “our cryo-ET model shows 34-fold symmetry for both the CCW and CW rotating C-rings”. Full details, ideally including images, on how symmetry was assigned must be provided and evidenced in the manuscript. This is especially important in light of the previous data showing variation in C-ring symmetry as, in the extreme, the relatively subtle differences between the CW and CCW volumes could reflect different proportions of 34mer in the samples being averaged rather than true differences between the states.

We agree with the reviewers that the symmetry of the C-ring is important for this study. Therefore, we have provided the detailed steps as requested by the reviewer 1. We first used Volta phase plate to increase the contrast of each reconstruction. We then used multivariate statistical analysis for 3D classification, which allowed us to resolve 34-fold symmetric features of the C-ring. Two independent approaches were used to determine the structures of the C-ring in both states. The resulting structures of the C-ring are considerably different.

3) Fitting of coordinates – it is not possible to judge the quality and uniqueness of the fit of coordinates to the volumes in the current version of the manuscript. A single correlation coefficient statistic is given in the text for each volume and it is unclear (1) if this is for the full model or simply the FliM_N_ spiral and (2) how each domain of the model contributes to this statistic. The figures that do show the fit show substantial extrusion of secondary structural elements outside of the volume and unfilled regions within the volume. This is perhaps unsurprising for a volume at modest resolution, but given that the authors go on to propose a radical structural rearrangement and then discuss the location of specific amino acids, the evidence for the uniqueness of the fit needs to be provided. Inspection of the volumes did not trivially allow the viewer to see a correlation in shape between the densities ascribed to the crucial FliG domains in the figures. There is also some confusion surrounding the FliM_c_/FliN spiral, as the text states that this portion of the structure is largely unaltered between the two states, but Figure 2 and Figure 2—figure supplement 1 show significant differences in the side views, with what looks like a 90 degree rotation of the spiral. How is this justified given the relatively featureless resolution of the maps?

We understand how it is very challenging to judge the placement of fitted, pseudoatomic models. However, we believe the model was necessary to help explain the cryo-ET maps. To address this issue, we did three things:

1) We made a supplemental figure of in silico maps reconstructed from the model at the resolution of our experimental maps. This was to show the level of detail that we should expect to see. The maps were similar to our experimental maps, but allow the reader to visualize what is different. We think that this will help with the idea of what can be interpreted from our pseudoatomic model at our resolution.

2) We clarified in the text that the amino acids are only talked about in relative location to help confirm the location of the proteins.

“It is important to note that these residues were used to identify unique locations in our model, but due to limited resolution their precise location is not known”

3) We made it clear that the spiral is not altered specifically in shape.

“This region, known as a spiral, like that of a spiral bound notebook, is uniform within both the CCW and CW cryo-ET maps, and remains largely unaltered upon rotational switching (Figure 4C). […] By building the patch model initially we were able to preserve the similarity of the MCN3 spiral by removing potential bias from the segmentation of a monomer. […] It is important to note that there may be slight movements of the MCN3 proteins during switching, that we would need higher resolution to observe.”

4) Discussion of novelty throughout should acknowledge the incremental nature of the current findings c.f. the earlier work. In particular the extensive discussion c.f. the work of Sakai should acknowledge that, at the resolution of both studies, both could be summarized as “no major difference in diameter on switching”.

We agree with the reviewers that our finding is an important extension of earlier work from many different labs. We have revised our manuscript with extensive discussions of other related studies.

5) CheY binding – this is the most speculative part of the manuscript as a ring of density around the CCW ring is assigned as CheY-P largely on the basis of the groups work that will be presented in another manuscript (cited as Chang et al. in preparation). Particularly in light of another manuscript recently reported in bioRxiv that demonstrates the N-terminus of FliM is not required for CheY-P binding induced switching (Eisenback et al.), the authors should either provide further supporting experiments or more clearly mark this as speculative.

We agree with the reviewers that this is speculative. Further experiments will be needed to support the model.

Reviewer #4:The paper presents in situ structures of the bacterial flagellar motor in CW and CCW states, and a corresponding detailed ("pseudoatomic") structural model of each state. Models are consistent with previous biochemical data on the proximity of various amino acid residues of the constituent proteins.The work is novel and important, and of sufficient general interest for publication in eLife: it allows analysis of the structural nature of the flagellar switch, which has been a canonical example of a macromolecular switch since the discovery of flagellar switching in the 1970s.1) There is a lack of clarity, perhaps even an apparent internal contradiction, as to whether the structure at the filament-distal end of the motor (the FliM_C_/ FliN_3_ super-helix, hereafter "MCN3 helix") is different between CW and CCW structures. This is an important point, worthy even of inclusion in the Abstract, as any changes in this part of the C-ring caused by the 2 point mutations in FliG would indicate long-range effects of very small mutational changes. Specifically: the MCN3 helix seems to be very different between CW and CCW in Figures 2C/D and Figure 2—figure supplement 1B/D – with fairly substantial rearrangement of protein chains. But the density maps for CW and CCW in Figure 2 A/B look very similar, and the lack of changes in the MCN3 helix is confirmed in Figure 4 and in the text. Is it possible that the repeating structure of 34 units has been cut differently between Figures 2C and D, and also Figures 2—figure supplement 1B and D? Or alternatively, that the outwardly similar shape of the MCN3 helix conceals a very different internal arrangement of protein chains? Or something else? This apparent contradiction needs to be addressed and removed. A pseudoatomic model allows and demands this level of inspection, with appropriate caveats about the confidence in the model itself.

We thank the reviewer for the critical concern. Our two structures of the C-ring suggested that the base of the C-ring remains in similar conformation. We therefore built the model based on the crystal structure of the FliN homolog (in McDowell et al., 2016). Our models were previously refined based on the segmented maps, which were unfortunately not reliable. Given the concerns and suggestions from the reviewer, we have used a different strategy, where we built a patch of 4 subunits, to refine the models. We believe that current models are consistent with the maps with 34-fold symmetry.

2) Closely related to comment 1, it is hard to match corresponding parts of the CW and CCW models – particularly in Figures 2C/D and Figure 2—figure supplement 1B/D. In reality there must be a pathway for switching between the CW and CCW structures. The authors should explore the transition between the two, and supply one or more animations to illustrate the conclusions that they can draw from their model regarding this important point – which is the main result of the paper. Such animations would be a powerful illustration, clarification and explanation of Figures 2-4 (and perhaps also Figure 5). They would serve 3 important purposes:a) Identify corresponding parts, including perhaps the possibility that the repeating structure has been cut differently to isolate the units displayed in Figure 2 (see comment 1).b) Illustrate either a smooth pathway between CW and CCW states (i.e. requiring no significant unfolding of secondary structure to avoid steric clashes, as proposed in the FliG model of Lee et al., 2010); or the lack of one, indicating rather the need for substantial disruption of interactions during a switch. The current best model for the switch is that of conformational spread (e.g. Bai et al., Science 2010), in which an energetic penalty for boundaries between domains of CW and CCW structure is a key parameter. This paper might be able to add structural data to that model.c) Highlight and illustrate changes in the orientation and location of FliG_c_, which are the "output" of the switch and most relevant in terms of understanding how it can cause a change in rotation direction upon interaction with the stator. These are discussed in the text, but it is hard to see them in the existing figures.

We agree and have made appropriate animations.

Video 1 is a summary of the project. Starting with the low resolution CCW map and moving to the locally refined C-ring map. Then building our pseudo-atomic model. We should how the symmetrized model switches between the CCW- and CW conformations by a simple toggle. And then move back out to the CW rotation.

Video 2 is a simple look at switching from above the C-ring. This shows how the charged residues of FliG change position relative to the stator if the stator were resolved in our maps.

Video 3 provides detail at how switching may occur in the domain swapping hypothesis. Here with zoom into a patch of the motor and watch how the CCW subunits morph into the CW subunits.

3) Regarding the extra density peripheral to the C-ring in the CW state, visible in Figure 1 and suggested to be CheY later in the paper (Figure 2—figure supplement 1, subsection “Extra diffuse density around the CW C-ring”). The authors should cite references that proposed that FliM_N_ is an unstructured "fishing rod" that serves to bind and concentrate CheY, which their hypothesis and data support. Also the alternative explanation (see e.g. Figure 7 in J Mol Biol. 430(22):4557-4579.) that this density could be extra copies of FliM, outside the ordered C-ring. This would reconcile previous observations of substantial recruitment of FliM to the C-ring (although in E. coli) during CW rotation with the lack of expansion of the 34-fold ring observed here.

We agree that our data alone does not identify the density as CheY-P, although it was recently suggested in two different studies. Alternatively, it could be also formed by extra FliM or FliN as suggested by the reviewer.

4) Figure 5 legend is wrong – D and C are described as if they were parts of B and A respectively. More importantly, the figure presents a model where the domain-swap (visible in B as swapping of colour-labelled domains of each protein chain into the neighbouring structural unit) is present in one directional state (CCW, B) but not the other (CW, D). This may be a misunderstanding of the domain-swap model of the flagellar switch. Alteration of the domain-swap during switching is not necessary, nor is it proposed by Baker et al., 2016 (as incorrectly stated in the Introduction). In fact, quite the opposite: the model proposes that the ARM-ARM interface is very strong, the main interaction holding the FliG ring together, and therefore very unlikely to open except under the special conditions that prevail during assembly of the FliG ring. Instead, the cartoon of the CCW state, Figure 5B, appears entirely consistent with the model of Lee et al., 2010, and the switch to a domain-swapped alternative of Figure 5D would then be a data-supported refinement and confirmation of the model of Lee et al., 2010. Alternative models propose non-domain swapped structures (like the cartoon of Figure 5D) for both CW and CCW. Perhaps the authors could comment on whether their data support one or other model.

Our resolution doesn’t not allow for the conformation of either model. As we push to increase resolution of the C-ring in tomography, hopefully we will be able to confirm via a true atomic resolution model. We built our models following the literature as closely as possible. Since many papers have suggested FliG is extended in the CCW rotation we used the full length FliG structure, 3HJL. To build this into the map we used the crystal packing that led to the domain swapping hypothesis. This FliG monomer assembled due to crystal packing fits into the map with little movement necessary. To build the CW model we used the “compact” FliG that has been proposed during CW rotation from the crystal structure 4FHR. The figures have been updated to clarify this by alternating the coloring of FliG. We have also addressed this in the text. We think that this is a piece of evidence to support this model, but it is not direct evidence.

We have updated our previous miscitation. We greatly thank the reviewer for noticing this.

“The domain swapping mechainism was proposed to coordinate with the conformational change in Helix_MC_ (Lee et al., 2010). Another biochemical study suggested that the domain swapping occurs during C-ring assembly, with FliG in solution existing as a monomer, and an equilibrium favoring FliG oligomers in the C-ring (Baker et al., 2016).”

[Editors’ note: what follows is the authors’ response to the second round of review.]

The manuscript has been improved but there are is a remaining issue that needs to be addressed before acceptance, as outlined below:In general, the authors have addressed the concerns raised in the previous review. But the decision letter for the previous manuscript stated: "A further concern was the preprint of Chang et al. recently posted on bioRxiv. Is it possible that the mechanism proposed in Chang et al. only applies to Borrelia burgdorferi, while the mechanism in the present paper only applies to Vibrio alginolyticus? If this is true, that should have been stated in both papers, but it is stated in neither." This is not answered either in the response nor in the revised paper.

First, we would like to thank the editors and the reviewers for your time and consideration of our revised manuscript. To some extent, the results in the preprint of Chang et al. matches with what we observed in *Vibrio alginolyticus*. We have updated our manuscript to make this clearer, although we had a sentence in the Discussion that may be overlooked. We replaced, “This finding is consistent with another structural characterization of the CW/CCW motors in *B. burgdorferi* (Chang et al., 2020).” with:

“In a parallel study, Chang et al. used a *B. burgdorferi* with a CheX deletion or CheY3 deletion to investigate the switching mechanism in spirochetes (Chang et al., 2020). We observed similar conformational changes of the C-ring in both species; however, the stator complexes were well resolved in *B. burgdorferi,* as they appear to be less dynamic than those in *V. alginolyticus.* Visualization of the stator complex provided direct evidence that the conformational change of the C-ring alters the rotor-stator interaction. The model of stator-rotor interactions controlling the rotational direction of the motor is further bolstered by the recent high-resolution cryo-EM structure of MotAB. Santiveri et al. further postulate that MotA rotates counterclockwise in *Campylobacter jejunie* by showing the conformations of wildtype MotAB and MotAB trapped in the protonated state, are nearly identical, thus, ruling out a dramatic conformational change in the stator unit.”

To further elaborate, the two manuscripts fit nicely together. We used different but analogous approaches to end up at the similar model in which the C-ring underwent a comparable conformational change in both species in different rotational directions. In *V. alginolyticus* we were able to generate single point mutation variants that locked the motors in CW or CCW rotation. This enabled us to be truly confident in the rotational direction of the flagellum motor. In *B. burgdorferi* we created motors that were half CCW by deleting CheY3, or all CW by deleting CheX, the kinase responsible for phosphorylation of CheY3. Spirochetes have a unique mode of movement as the flagella are located at both cell poles, therefore during the “run” one cell pole is rotating CCW and the other rotating CW (potentially due to an unknown protein). In *B. burgdorferi* we were able to resolve density for the stator complex, this has been shown before but not to such resolution. The ability to resolve the stator helps identify FliG-MotA interactions. We are working on trying to visualize the stator in the *V. alginolyticus* motors, but the stator occupancy seems to be more dynamic making this a very challenging task. The *B. burgdorferi* visualization of the stator, reinforced by the new stator cryo-EM structures recently put on bioRxiv, provides additional evidence for our model. We were unable to discuss much about the stator involvement in this manuscript. But comparison of our CCW and CW C-rings revealed an unambiguously similar pattern with the movement of the C-ring repositioning FliGC thus altering the FliG-MotA interaction.